# The making of a potent L-lactate transport inhibitor

Patrick D. Bosshart[1,2], David Kalbermatter [1,3], Sara Bonetti[1,3] & Dimitrios Fotiadis [1✉]

L-lactate is an important metabolite, energy source, and signaling molecule in health and disease. In mammals, its transport across biological membranes is mediated by mono-carboxylate transporters (MCTs) of the solute carrier 16 (SLC16) family. Malfunction, overexpression or absence of transporters of this family are associated with diseases such as cancer and type 2 diabetes. Moreover, lactate acts as a signaling molecule and virulence factor in certain bacterial infections. Here, we report the rational, structure-guided identification of potent, nanomolar affinity inhibitors acting on an L-lactate-specific SLC16 homologue from the bacterium *Syntrophobacter fumaroxidans* (SfMCT). High-resolution crystal structures of SfMCT with bound inhibitors uncovered their interaction mechanism on an atomic level and the role of water molecules in inhibitor binding. The presented systematic approach is a valuable procedure for the identification of L-lactate transport inhibitors. Furthermore, identified inhibitors represent potential tool compounds to interfere with mono-carboxylate transport across biological membranes mediated by MCTs.

[1] Institute of Biochemistry and Molecular Medicine, and Swiss National Centre of Competence in Research (NCCR) TransCure, University of Bern, Bern, Switzerland. [2] Present address: leadXpro AG, Park Innovare, Villigen, Switzerland. [3] These authors contributed equally: David Kalbermatter, Sara Bonetti. ✉email: dimitrios.fotiadis@ibmm.unibe.ch

The solute carrier 16 (SLC16) family constitutes a diverse group of membrane transporters composed of 14 genes in the human genome (i.e., *SLC16A1-A14*)[1]. The broad range of transported substrates (e.g., L-lactate, pyruvate, ketone bodies, short-chain fatty acids, thyroid hormones, aromatic amino acids, amino acid metabolites, or drugs) reflects the diversity of the SLC16 family[1]. Within the transporter classification (TC) system, the SLC16 family represents the monocarboxylate transporter (MCT) family (TC-ID 2.A.1.13), which is a subgroup of the major facilitator superfamily (MFS, TC-ID 2.A.1)[2]. MFS transporters share a canonical fold, which is characterized by 12 transmembrane helices (TMs) assembled into two interconnected six-helix bundles[3]. These bundles are related to each other by a pseudo-twofold symmetry axis that runs perpendicular to the membrane plane and lies in the substrate translocation pathway. An arginine residue is conserved in transmembrane helix 8 of most SLC16 family members and its positively-charged guanidinium group is involved in the binding of the negatively-charged carboxylate group of substrates and ligands[4–6]. Several SLC16 family members are associated with health disorders. For example, MCT1 and MCT4 are overexpressed in certain tumors and play an essential role in their metabolism[7]. MCT11 and MCT13 are associated with type 2 diabetes risk[8,9]. Mutations in the *MCT8* gene are connected to the Allan-Herndon-Dudley syndrome[10], whereas a mutated *MCT12* gene is linked to a syndrome, whose symptoms include juvenile cataract and glucosuria[11]. Furthermore, MCT1 deficiency is a cause of ketoacidosis[12] and its failed silencing causes exercise-induced hyperinsulinism[13].

L-lactate is considered an important glycolytic metabolite, energy source, and signaling molecule in human health and disease[14]. Additionally, lactate has an important role in some bacterial infections. In *Neisseria meningitidis*, it acts as a signaling molecule for microcolony dispersal[15]. In *Staphylococcus aureus*-induced biofilms, it functions as a virulence factor, sustaining an anti-inflammatory environment that promotes bacterial persistence[16].

We have recently determined crystal structures of the proton-dependent, L-lactate-specific SLC16 family homologue SfMCT with bound L-lactate and thiosalicylate (TSA), which is a non-transported ligand[6]. The structures show SfMCT in the pharmacologically-relevant outward-open conformation and they provide important mechanistic insights into the role of critical residues involved in ligand binding[6]. The negatively-charged carboxylate groups of L-lactate and TSA interact with the positively-charged guanidinium group of R280 (TM8), which is conserved in TM8 of most SLC16 family members[17]. Here, we use this structural information for the rational identification of potent, nanomolar (nM) affinity L-lactate transport inhibitors starting from a low-affinity compound. We have solved high-resolution crystal structures of SfMCT with bound potent inhibitors to understand the molecular mechanism underlying the binding of these molecules. Furthermore, we uncovered the role of water molecules in the interaction between inhibitors and key residues based on high-quality structural data. Identified inhibitors represent potential tool compounds to interfere with monocarboxylate transport across biological membranes.

## Results
### Rational identification of potent L-lactate transport inhibitors.
The thiosalicylate (**TSA**)-bound structure of the proton-dependent, L-lactate-specific SLC16 homologue SfMCT in its outward-open conformation served as a structural framework for the rational identification of potent L-lactate transport inhibitors[6]. The previously characterized ligand binding site provides space to accommodate molecules that are larger than **TSA**. Since **TSA** is a sulfanyl-

derivative of the aromatic monocarboxylate benzoate (**BA**), **BA** was selected as starting compound of a rational search for molecules that potently inhibit L-lactate transport (Supplementary Fig. 1). Only molecules containing one carboxylate-group were considered (i.e., monocarboxylates) as R280 (TM8) is the only positively-charged residue in the ligand binding site region. Furthermore, previous experiments have shown that SfMCT does not have significant affinity for di- and tricarboxylates[6]. The binding site also contains hydrophobic and aromatic residues, which can be involved in hydrophobic and π-stacking interactions with bound molecules. Therefore, the focus of the screen was set on aromatic monocarboxylates, which are summarized in Supplementary Fig. 1. To evaluate the inhibitory effect of the selected molecules, an inhibition assay was set up where the transport of [$^{14}$C]L-lactate through SfMCT was measured in the absence and presence of different inhibitors at decreasing concentrations. In a first step, the inhibitory effects of **BA**, as well as hydroxylated and methylated **BA** derivatives on L-lactate transport, were investigated at a concentration of 500 μM (Fig. 1A, red). Among the hydroxylated **BA** derivatives, a hydroxyl group at the C2-carbon atom of **BA** (i.e., salicylate (**SA**)) leads to the strongest decrease in L-lactate transport activity. **BA** and **SA** inhibit L-lactate transport with comparable $K_i$ values of 431 μM and 362 μM, respectively (Fig. 1B, C). In contrast to hydroxylation, the C4-carbon atom of **BA** is the optimal position for derivatization by a methyl-group (i.e., 4-methyl-BA (**4MBA**)) resulting in the strongest reduction of L-lactate transport activity followed by methylation at the C3- and C2-carbon atoms (Fig. 1A, red). **4MBA** inhibits L-lactate transport with a $K_i$ of 13.4 μM (Fig. 1D), which is ~30 times more potent than **BA**, the starting compound of the rational search. Inhibition experiments performed at a compound concentration of 100 μM show that combining hydroxylation and methylation of **BA** at the determined optimal positions (i.e., C2-carbon atom for hydroxylation, C4-carbon atom for methylation) does not lead to a synergistic improvement of L-lactate transport inhibition (Fig. 1A, blue). As observed for **BA** derivatives, methylation at the C4-carbon atom of **SA** (4-methyl-SA (**4MSA**)) leads to the strongest reduction of L-lactate transport among the methylated **SA**-derivatives with a $K_i$ of 24.4 μM (Fig. 1E). This inhibitory potency is comparable with the value obtained for **4MBA**. The hydroxyl group of **4MSA** cannot be shifted from the C2- to the C3-carbon atom as shown by the strongly reduced inhibition of L-lactate transport by 3-hydroxy-4-methyl-BA (**3OH4MBA**; Fig. 1A, blue). The fact that adding methyl-groups to the C3- and C4-carbon atoms increased the inhibitory effect of **BA** and **SA**, suggests fusing a second benzene ring to these compounds to further increase their affinity. This results in two naphthoates (i.e., 1-naphthoate (**N1C**) and 2-naphthoate (**N2C**)), and three hydroxy-naphthoate derivatives (i.e., 1-hydroxy-2-naphthoate (**1OHN2C**), 3-hydroxy-2-naphthoate (**3OHN2C**) and 2-hydroxy-1-naphthoate (**2OHN1C**)). The inhibitory effect of these compounds on L-lactate transport was determined at a concentration of 25 μM (Fig. 1A, orange). **N2C**, where the second benzene ring is fused to **BA** via the C3- and C4-carbon atoms, and **1OHN2C** show the strongest reduction of L-lactate transport among the tested naphthoate derivatives with $K_i$ values of 2.4 and 16.2 μM, respectively (Fig. 1F, G). **1OHN2C** revealed a comparable $K_i$ value as observed for **4MBA** and **4MSA**. The naphthalene rings of **N2C** and **1OHN2C** are completely planar. In order to increase conformational flexibility, the carboxylate-containing benzene ring of **N2C** was replaced by an $sp^2$ hybridized alkene (i.e., trans-cinnamate (**CA**)) or an $sp^3$ hybridized alkane (i.e., 3-phenyl-propionate (**3PP**)) chain. **CA** and **N2C** have similar $K_i$ values of 2.5 and 2.4 μM, respectively (Fig. 1F, H). In contrast, **3PP** shows a stronger inhibitory effect compared to **N2C** (Fig. 1A, gray), which is highlighted by a lower $K_i$ value of 812 nM (Fig. 1I). Accordingly, we selected aromatic monocarboxylates where the

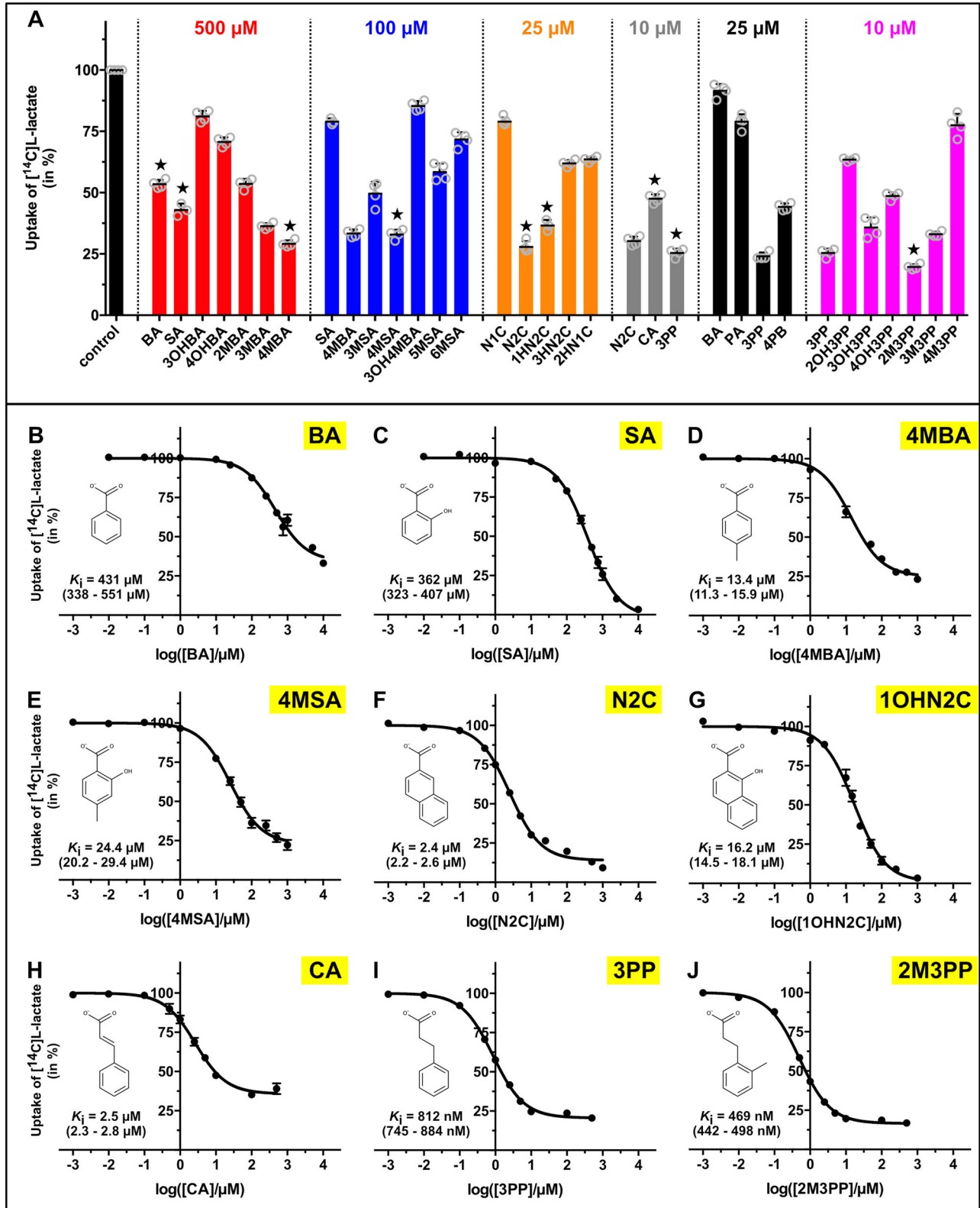

**Fig. 1 Identification of SfMCT inhibitors. A** Screening for L-lactate transport inhibitors using a transport inhibition assay (final inhibitor concentrations are indicated). Full names and molecular structures of the used compounds are given in Supplementary Fig. 1. For compounds highlighted by a *star*, $K_i$ values were determined. Residual uptake in the presence of competitor is normalized with respect to control samples without competitor (control). **B**–**J** $K_i$ determination of selected inhibitors with displayed structures. The determined $K_i$ values and the 95% confidence intervals are indicated in the corresponding panels. Data are represented as mean ± SEM from three to five independent experiments, each in triplicate. If not visible, error bars are smaller than symbols. Individual data points are shown as open circles.

carboxylate-group and the benzene ring are connected by an aliphatic chain in the further rational screening procedure. Aromatic monocarboxylates of different length (i.e., benzoate (**BA**), phenyl-lacetate (**PA**), 3-phenylpropionate (**3PP**), and 4-phenylbutyrate (**4PB**); Fig. 1A, black) served as molecular rulers to determine the optimal length of the aliphatic linker. Among the tested compounds, **3PP** is the most potent inhibitor (Fig. 1A, black), which is ~530 times more potent than the starting compound (**BA**) of the rational search. In a final step, the effect on L-lactate transport of hydroxylation and methylation of the benzene ring of **3PP** was systematically analyzed at an inhibitor concentration of 10 μM (Fig. 1A, magenta). Hydroxylation of the C3-carbon atom (3-(3-hydroxyphenyl)-propionate (**3OH3PP**)) leads to the strongest L-lactate transport reduction among the tested hydroxylated derivatives (Fig. 1A, magenta). In contrast, a methyl-group at the C2-carbon atom (3-(2-methylphenyl)-propionate (**2M3PP**)) reveals the strongest inhibitory effect on L-lactate transport among the tested methylated derivatives. **2M3PP** inhibits L-lactate transport with a $K_i$ value of 469 nM (Fig. 1J), which is significantly lower than the inhibition induced by **3PP**. In contrast to SfMCT, the here identi-fied key inhibitors (i.e., **1OHN2C**, **N2C**, **3PP**, and **2M3PP**) do not affect the transport activity of *E. coli* L-lactate permeases (i.e., LldP and GlcA), whose function depends on an intact transmembrane proton-gradient (Supplementary Fig. 2)[18]. Therefore, the reduced SfMCT-mediated L-lactate transport, which was measured in the presence of these key inhibitors, is due to transport inhibition and does not result from a protonophoric effect of these molecules. We have also performed a transport assay using radiolabeled **SA** and **3PP**, which are both key compounds of our rational search. SfMCT does not transport **SA** or **3PP**, which allows the assumption that also derivatives thereof are not transported (Supplementary Fig. 3). In summary, the presented rational screening approach allowed the identification of two nM-affinity inhibitors starting from a low-affinity compound (i.e., **BA**). The best inhibitor **2M3PP**, which has a $K_i$ value of 469 nM, is almost 1000 times more potent than the starting compound of the rational search.

**Crystal structures of SfMCT with bound inhibitors**. To understand the molecular interactions underlying the binding of the identified key inhibitors (i.e., **1OHN2C**, **N2C**, **3PP**, and **2M3PP**), we established a procedure to successfully co-crystallize SfMCT with these compounds. Purified and concentrated SfMCT (8 mg/ml; ~0.18 mM) was supplemented with 1 mM of key inhibitors before setting up sitting-drop crystallization trials. Crystals of sufficient size for high-quality data collection only grew in the presence of inhibitors. This allowed us to solve the crystal structures of SfMCT with bound inhibitors at resolutions of 2.39–2.67 Å (Supplementary Table 1). For all inhibitors, the transporter adopts the pharmacologically-relevant outward-open conformation, where the previously characterized binding site is accessible from the extracellular side (Fig. 2A)[6]. In all cases, the binding site is occupied by one inhibitor molecule (Fig. 2B–F; for OMIT maps, see Supplementary Fig. 4). Residues from the amino- and carboxy-terminal six-helix bundles interact through ionic (R280 (TM8)), hydrogen bond (Y331 (TM10), Y119 (TM4)), hydrophobic (L28 (TM1), C57 (TM2), F60 (TM2), L145 (TM5), F335 (TM10), F359 (TM11), C362 (TM11)) and π-stacking (F359 (TM11)) interactions with the inhibitors (Figs. 2 and 3). Furthermore, water molecules adopt important roles in inhibitor binding. As previously observed for L-lactate and **TSA**, the carboxylate-groups of all inhibitors are positioned in an ~8 Å wide confinement formed by the side chains of L145 (TM5) and F335 (TM10) (Fig. 3)[6]. Both oxygen atoms of the negatively-charged carboxylate-group of all inhibitors form a salt-bridge with the Nη1 and Nη2 nitrogen atoms of the positively-charged guanidinium

group of the conserved and functionally important R280 (TM8). The carboxylate-groups of the bound inhibitors also interact with the hydroxyl group of Y119 (TM4) via hydrogen bonding. The orientation between the guanidinium plane of R280 (TM8) and the carboxylate-groups of bound key inhibitors was quantified by measuring their interaction geometry (Fig. 2G, H; Supplementary Fig. 5). In all cases, the angles between the guanidinium and the carboxylate planes are in a range that is commonly observed for arginine-carboxylate interactions[19]. In contrast, **TSA** interacts only with one oxygen atom of its carboxylate-group with the Nη atoms of R280 (TM8) (Fig. 2F; Supplementary Fig. 5). Further-more, the interaction geometry is not in the range that is com-monly observed for arginine-carboxylate interactions[19]. The naphthalene ring of **1OHN2C** is almost in plane with its car-boxylate-group, which can be attributed to an intramolecular hydrogen bond between the hydroxyl and carboxylate groups[20]. The hydroxyl group of **1OHN2C** is in hydrogen bonding distance to the hydroxyl group of Y331 (TM10), which plays a role in ligand recognition[6]. In contrast to **1OHN2C**, the naphthalene ring of **N2C** is rotated by ~40° with respect to the carboxylate-group. As observed for **N2C**, the benzene rings of **3PP** and **2M3PP** are also not in plane with their carboxylate-groups. The rotation of the benzene or naphthalene rings with respect to the carboxylate groups is associated with a higher conformational flexibility. This allows for a better adaptation to the binding site of SfMCT and seems to be important for improving the affinity of inhibitors.

**Role of Y331 and water molecules in ligand binding**. An aro-matic residue in TM10, which corresponds to Y331 in SfMCT, is involved in ligand recognition in several bacterial and human L-lactate transporters[5,6]. While the hydroxyl group of Y331 (TM10) is directly hydrogen-bonded to the hydroxyl group of **1OHN2C** and the thiol group of **TSA**, it interacts indirectly via a water molecule with the carboxylate groups of **N2C**, **3PP**, and **2M3PP** (Fig. 2B–F). Therefore, the hydroxyl and thiol groups as well as the identified water molecule are important for the interaction with the functionally-relevant Y331 (TM10) and thus have a similar functional role in ligand binding. In addition to structural data, transport inhibition experiments corroborate the important role of Y331 (TM10) in ligand recognition. Replacing Y331 (TM10) by a phenylalanine, which removes the hydroxyl group of Y331 (TM10), results in a ~2-fold reduction of L-lactate transport inhibition by **1OHN2C**, **N2C**, **3PP**, and **2M3PP** (Supplementary Fig. 6). This highlights the functional significance of the hydroxyl group of Y331 (TM10) for ligand recognition. A second water molecule was identified as part of an extended hydrogen-bonding relay network, which connects functionally-relevant residues from the amino- and carboxy-terminal six-helix bundle (i.e., L145 (TM1), N276 (TM8), R280 (TM8), and Y331 (TM10); Supplementary Fig. 7)[6].

**Hydrophobic interactions**. In addition to salt bridges and hydrogen bonds, hydrophobic interactions contribute to inhibitor binding (Fig. 3). The benzene ring of F335 (TM10), which is part of the carboxylate group-accommodating confinement, is involved in a hydrophobic interaction with the aromatic moiety of naphthoate-derivatives, as well as with the alkane chains of **3PP** and **2M3PP** (Fig. 3). Furthermore, there is a hydrophobic interaction between the methyl-group of **2M3PP** and the aro-matic side chain of F335 (TM10), which provides an additional binding interaction with SfMCT compared to **3PP**. The surface representation of the binding site region clearly highlights that the methyl-group of **2M3PP** points towards a cavity between TM7 and TM10 (Fig. 3H). This cavity is exclusively located in the carboxy-terminal six-helix bundle and therefore dictates the

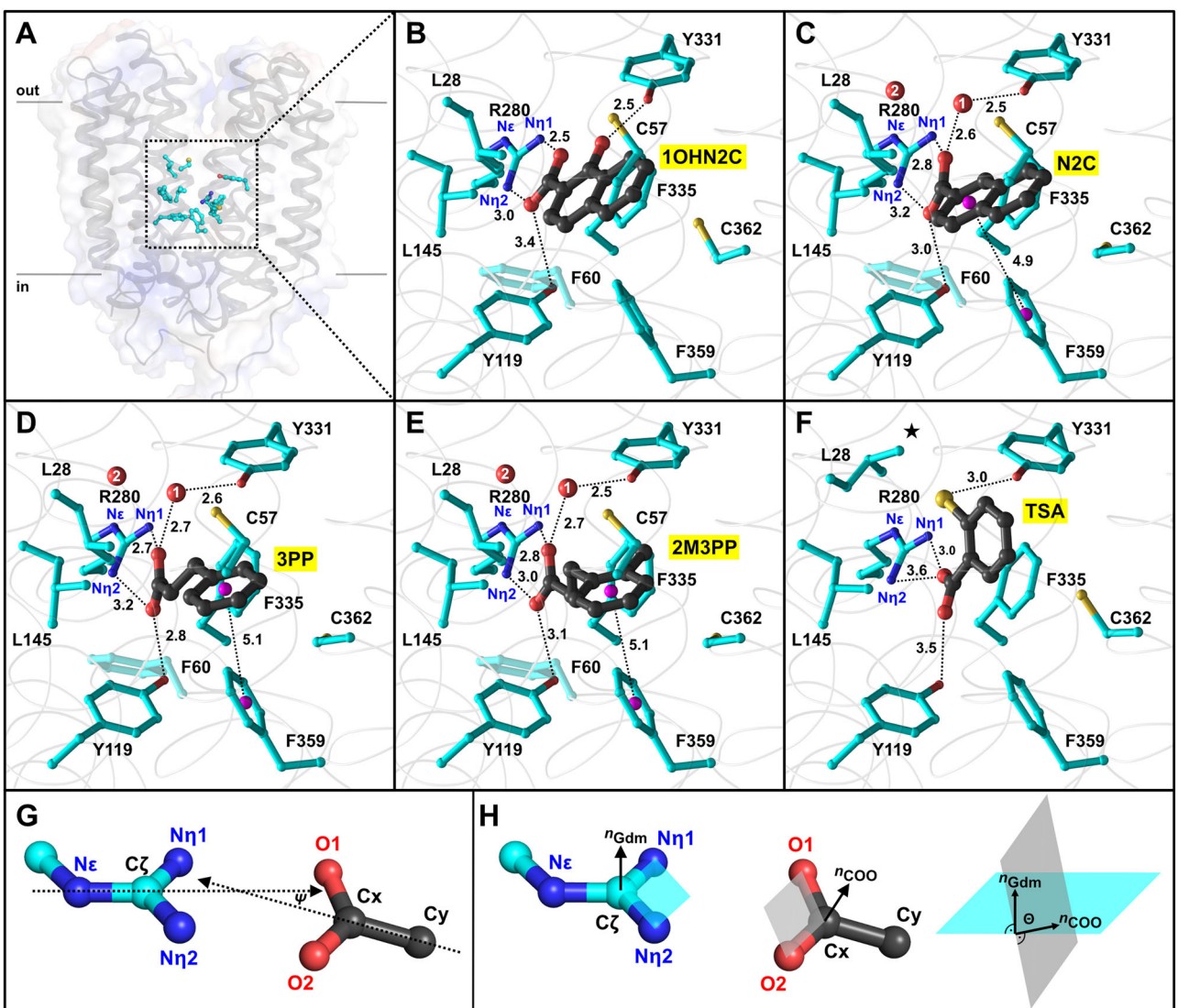

**Fig. 2 SfMCT ligand binding site. A** Overall structure of SfMCT in the outward-open conformation viewed in the plane of the membrane with indicated binding site residues (cyan) and surface representation. **B–F** Binding mechanism of **1OHN2C**, **N2C**, **3PP**, **2M3PP**, and **TSA** to SfMCT. Corresponding omit maps are shown in Supplementary Fig. 4. Residues within a distance of 4 Å from the bound compounds are displayed as ball-and-stick models and highlighted in cyan. Pink spheres indicate the centers of benzene rings of aromatic residues and inhibitors, which are involved in π–π stacking interactions. Water molecules are indicated by labeled red spheres. The role of the second water molecule (2) is shown in Supplementary Fig. 7. The different rotamer conformation of L28 in the case of TSA binding is indicated by a star in (**F**). Distances are given in Ångström (Å). **G**, **H** The orientation between the guanidinium plane of R280 and the carboxylate-groups of bound inhibitors can be quantified by measuring the angle $\psi$ between the Nε–Cζ and Cx–Cy bonds as well as the angle $\theta$ between the guanidinium (cyan) and the carboxylate (gray) planes. The following angles were measured: **1OHN2C** $\psi = 15°$, $\theta = 15°$; **N2C** $\psi = 16°$, $\theta = 20°$; **3PP** $\psi = 17°$, $\theta = 31°$; **2M3PP** $\psi = 11°$, $\theta = 18°$; **TSA** $\psi = 69°$, $\theta = 16°$. PDB IDs of displayed structures are 6ZGR (**1OHN2C**), 6ZGS (**3PP**), 6ZGT (**N2C**), 6ZGU (**2M3PP**) and 6G9X (**TSA**).

orientation of bound **2M3PP**. A hydrophobic environment formed by L28 (TM1), C57 (TM2), F60 (TM2), F359 (TM11), and C362 (TM11) accommodates the second benzene ring of the naphthalene moiety of **1OHN2C** and **N2C** as well as the benzene ring of **3PP** and **2M3PP** (Fig. 3). Structural analysis revealed that the carboxylate group-containing benzene ring of **N2C** and the benzene rings of **3PP** and **2M3PP** are involved in a T-shaped π-stacking interaction with the aromatic side chain of F359 (TM11) (Fig. 3)[21]. With the exception of L28 (TM1), residues, which are involved in inhibitor binding, adopt similar rotamer conformations in SfMCT crystal structures with bound key inhibitors and bound **TSA** (Fig. 2). In contrast to the **TSA**-bound structure, L28 (TM1) is rotated away from the central cavity in the presence of the here identified inhibitors (Fig. 2F, star). This rotamer

conformation of the isobutyl side-chain of L28 (TM1) allows the formation of hydrophobic interactions with the carboxylate group-containing benzene ring of the naphthoate-derivatives (Fig. 3A, B) or with the benzene ring of **3PP** and **2M3PP** (Fig. 3C, D). The distance between the benzene ring of **3PP** and the adjacent TMs 2 and 11 is <4.5 Å, which precludes the extension of the inhibitor by another methylene-group. This structural information is in line with functional data. In a molecular ruler experiment using aromatic monocarboxylates, which differ in the length of the aliphatic chain connecting the carboxylate-group and the benzene ring, as L-lactate transport inhibitors (i.e., **BA**, **PA**, **3PP**, and **4PB**; Fig. 1A), **3PP** was identified as the inhibitor with the optimal length among the tested molecules (Fig. 1A, black). Apparently, while longer molecules (i.e., **4PB**) do not

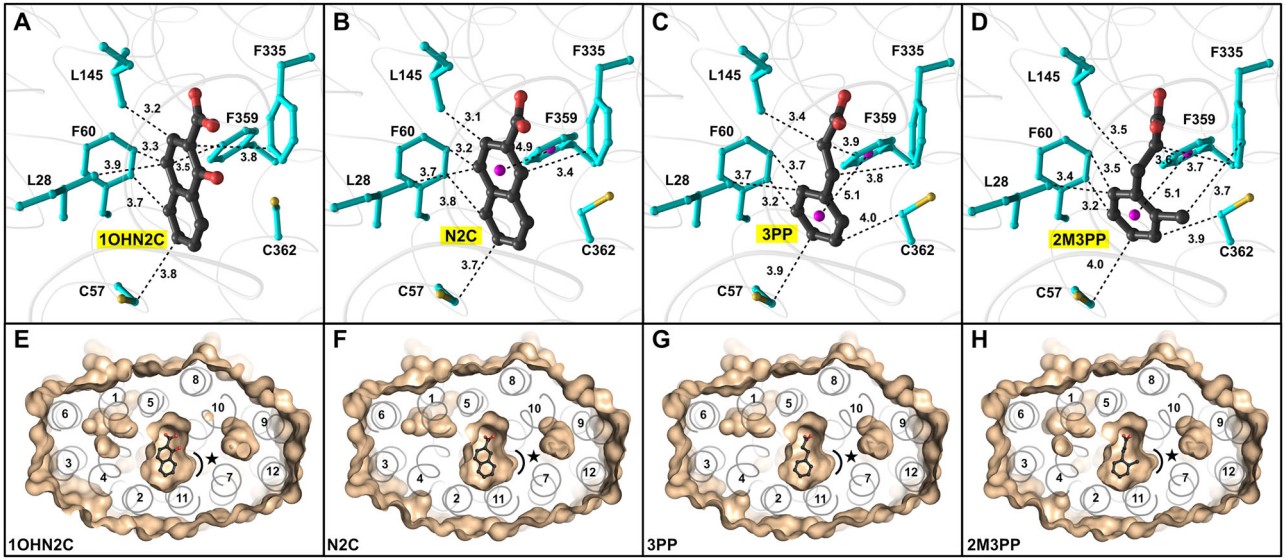

**Fig. 3 Hydrophobic interactions involved in ligand binding.** Ligands **A** 1OHN2C, **B** N2C, **C** 3PP, and **D** 2M3PP as determined by structural analysis[21]. Hydrophobic residues within a distance of 4 Å from the bound compounds are displayed as ball-and-stick models and highlighted in cyan. The centers of benzene rings of aromatic residues and inhibitors, which are involved in π–π stacking interactions, are indicated by pink spheres. Distances are given in Ångström (Å). **E–H** Surface representations of the inhibitor binding site region. Bound compounds are displayed as ball-and-stick models. A cavity between TMs 7 and 10 is indicated by a black curved line and a star. PDB IDs of displayed structures are 6ZGR (**1OHN2C**), 6ZGS (**3PP**), 6ZGT (**N2C**), and 6ZGU (**2M3PP**).

properly fit into the binding pocket, shorter compounds (e.g., **BA**, **PA**) cannot establish sufficient interactions with SfMCT, which reduces their affinities.

## Discussion

L-lactate is no longer solely regarded as glycolytic end-product, but is now considered an important energy source and signaling molecule in human health and disease[14]. Also, there is evidence of its role as a virulence factor and signaling molecule during bacterial infections[15,16]. In human, the transport of this negatively-charged monocarboxylate across the hydrophobic core of biological membranes is mediated by members of the SLC16 family, which are homologues of the here studied SfMCT. In pathogenic bacteria, the presence of SfMCT-related lactate transporters might help promoting infections[15,16]. Using a rational and structure-guided screening approach with a low-affinity starting compound, we have successfully identified a series of monocarboxylates that efficiently inhibit SfMCT-mediated L-lactate transport. The identified inhibitors **3PP** and **2M3PP** have $K_i$ values in the nM-range, which are up to ~1000 times lower than the one measured for the starting compound **BA** (Fig. 1B, I and J). The corresponding, inhibitor-bound SfMCT crystal structures highlight that the negatively-charged carboxylate groups of the inhibitors are bound to the positively-charged guanidinium group of R280 (TM8; Fig. 2) as also observed for the L-lactate- and **TSA**-bound structures[6]. The fact that the bound inhibitors share the same binding site as the transported substrate L-lactate suggests that these molecules act as competitive inhibitors. In the previously published L-lactate- and **TSA**-bound SfMCT crystal structures, only one oxygen atom of the carboxylate group interacted with the guanidinium group of R280 (TM8)[6]. This binding mode has also been observed in the substrate-bound crystal structure of the galactonate membrane transporter DgoT, which is an SLC17 family homologue[22]. In contrast, both carboxylate oxygen atoms of the here identified and co-crystallized inhibitors were involved in salt-bridges with this particular guanidinium group, which was also observed for carboxylate-group containing drugs bound to their targets[23–26]. These drugs had similar interaction geometries as determined for the here identified SfMCT inhibitors. The fact that the interaction geometries between

the carboxylate groups of SfMCT inhibitors and the guanidinium group of R280 (TM8) are within a preferred range might be the reason for their lower $K_i$ values compared to **TSA**[19]. In addition to a salt-bridge established to R280 (TM8), the carboxylate groups of the here identified inhibitors are also hydrogen-bonded to the hydroxyl group of Y119 (TM4). The combination of an interaction to an arginine and the hydroxyl group of a tyrosine was also observed for the binding of benzoate to an amino acid oxidase[23] as well as for the binding of carboxylate-containing drugs to cyclooxygenases[27–29]. Our structural and functional data highlight the critical role of Y331 (TM10) in ligand binding. Whereas Y331 (TM10) is directly hydrogen-bonded to the hydroxyl group of **1OHN2C** and the thiol group of **TSA**, it interacts indirectly via a water molecule with the carboxylate groups of **N2C**, **3PP**, and **2M3PP** (Supplementary Fig. 7). Comparison of residues that are important for inhibitor-binding in SfMCT (Figs. 2 and 3) with corresponding residues in the structure of human MCT1[30] indicates that there is no strong conservation except for the conserved and functionally-essential positively-charged arginine residue in TM8 (Supplementary Fig. 8). This residue is involved in ligand binding in prokaryotic and mammalian SLC16 transporters[6,30,31]. Alignment of the **2M3PP**-bound SfMCT structure with an outward-open cryo-EM structure of human MCT1[30] suggests that the here presented SfMCT inhibitors (i.e., **1OHN2C**, **N2C**, **3PP**, **2M3PP**) might be differently oriented in human MCT1 since the distance between the Nη atoms of the guanidinium group of R313 and the oxygen atoms of the carboxylate group of **2M3PP** are 4.7 and 4.9 Å (Supplementary Fig. 8C), which is significantly larger than observed for the inhibitor-bound SfMCT structure (Fig. 2). Furthermore, there would be clashes between the inhibitors and MCT1 side chains (Supplementary Fig. 8). Nevertheless, it can be hypothesized that SfMCT inhibitors might indicate transport-modulating effects on certain human MCTs as all compounds contain a carboxylate group.

Water molecules are involved in ligand binding in several membrane transporters where they bridge the distance between transporter residues and the bound ligands[32–38]. Removing the hydroxyl group of Y331 (TM10) (i.e., Y331F mutation) reduces the transporter affinity for the identified inhibitors

(Supplementary Fig. 6). This has also been previously shown for the binding of **TSA**[6]. A second water molecule is part of an extended hydrogen-bonding relay network, which connects functionally-relevant residues in the binding site region (Supplementary Fig. 7). It can be concluded that the identified water molecules contribute to the correct spatial positioning of side chains involved in ligand binding (i.e., L145 (TM5), N276 (TM8), R280 (TM8), Y331 (TM10)). The here presented SfMCT crystal structures provide a molecular explanation for the fact that **2M3PP** has the best inhibitory potency among the characterized compounds. First, the $sp^3$ hybridized alkane chain that connects the carboxylate group with the benzene ring allows a higher conformational flexibility of **3PP** and **2M3PP** compared to **1HN2C** and **N2C**, which is important for efficient inhibitor binding. Second, surface analysis of all inhibitor-bound crystal structures reveals an asymmetric binding site region with a cavity in the carboxy-terminal six-helix bundle between TM7 and TM10 (Fig. 3E–H, star). The methyl group of **2M3PP** points towards this cavity and dictates the orientation of bound **2M3PP**, since no cavity has been identified in the amino-terminal six-helix bundle. The compound **3PP** represents a promising inhibitor scaffold, whose length is mainly restricted by TMs 2, 8, and 11 (Fig. 3G). The **3PP**- and **2M3PP**-bound, outward-open SfMCT crystal structures suggest that the inhibitor size can be vertically increased in the direction of the substrate translocation pathway by attachments at the Cβ carbon atom (Supplementary Fig. 1).

In summary, we have identified nM-affinity transport inhibitors of the L-lactate-specific SLC16 homologue SfMCT using a rational, structure-guided screening approach, which started with a low-affinity compound. Accordingly, the inhibitory potency could be increased by almost three orders of magnitude. We have solved high-resolution crystal structures of SfMCT with bound inhibitors, which provided important insights into the binding mechanism of these molecules on an atomic level. These crystal structures clearly explain the molecular reasons, which underlie the improvement of the inhibitory potency of the identified inhibitors. This highlights the impact of high-resolution, inhibitor-bound crystal structures on the interpretation of inhibition data and the design of potent inhibitors. The here identified L-lactate transport inhibitors represent valuable compounds that might potentially also interfere with substrate transport across biological membranes mediated by SLC16 family members. In contrast to human MCTs, only a very low number of the innumerable bacterial homologs have been studied. Considering the presence of SLC16 family homologues in human pathogenic bacteria, application of monocarboxylate transport inhibitors to compromise pathogen's survival might turn out as a valid antibacterial approach in the near future.

## Methods

**Cloning of SfMCT**. Cloning of the bacterial SfMCT (UniProt ID code A0LNN5) gene into the pZUDF21-rbs-3C-10His plasmid was done as described previously[6,39,40]. The gene, which was codon-optimized for expression in *E. coli* (GenScript), was ligated into the pZUDF21-rbs-3C-10His plasmid using 5′-HindIII and 3′-XhoI restriction sites for overexpression, or into the pEXT20 plasmid using 5′-EcoRI and 3′-XhoI restriction sites for functional studies[41]. The resulting constructs (pZUDF21-rbs-SfMCT-3C-10His and pEXT20-SfMCT-3C-10His) contained recombinant SfMCT followed by a C-terminal human rhinovirus 3C (HRV3C) protease cleavage site and a decahistidine-tag (His-tag). The Y331F mutant was generated by QuikChange site-directed mutagenesis (Agilent Technologies) using the primer (5′–3′) GTT ATC GGT TGG AAC TTT GGC GCA ATG TTT ACC.

**Uptake of radiolabeled L-lactate into SfMCT-expressing *E. coli***. Uptake experiments were done as described previously[6,40]. Overnight precultures of *E. coli* JA202 (MC4100 *glcA::cat lldP::kan*)[42], which were transformed with a plasmid carrying SfMCT (pEXT20-SfMCT-3C-10His), were diluted 1:200 into LB-medium supplemented with 100 μg/ml ampicillin. Cultures were grown at 37 °C and 180 r.p.m. in an incubator shaker (Multitron, Infors HT). At OD₆₀₀ ~0.5, isopropyl-β-D-thiogalactopyranoside was added to the culture to a final concentration of 250 μM to induce SfMCT expression. In the case of *E. coli* MC4100 (DSM 6574, German Collection of Microorganisms and Cell Cultures), an overnight preculture was diluted 1:200 into LB-medium supplemented with 50 μg/ml streptomycin and grown at 37 °C and 180 r.p.m. in an incubator shaker (Multitron, Infors HT). Bacteria were harvested 4 h post-induction (*E. coli* JA202) or 6 h post-inoculation (*E. coli* MC4100) (5200 × *g*, 10 min, room temperature). The supernatant was discarded and the bacteria were gently resuspended in uptake buffer (20 mM Bis-Tris propane-HCl (pH 6.7), 250 mM KCl). The density of the bacteria suspension was adjusted to OD₆₀₀ 12. Transport experiments were done in a reaction volume of 50 μl, which consisted of 20 μl cell-suspension (2.4 × 10⁸ bacteria), 10 μl substrate master mix (67 μM sodium L-lactate spiked with [14C(U)] L-lactic acid sodium salt ([¹⁴C]L-lactate, American Radiolabeled Chemicals) to a specific activity of 0.15 Ci/mmol) and 20 μl monocarboxylate inhibitor solution. In the case of transport experiments using L-lactate, salicylate, and 3-phenylpropionate for comparison, the reaction volume of 50 μl consisted of 20 μl cell-suspension (2.4 × 10⁸ bacteria) and 30 μl substrate master mix (67 μM sodium L-lactate spiked with [14C(U)] L-lactic acid sodium salt ([¹⁴C]L-lactate, American Radiolabeled Chemicals), or 67 μM salicylic acid spiked with [7–14C] salicylic acid ([¹⁴C]salicylic acid, American Radiolabeled Chemicals), or 67 μM 3-phenyl propionic acid spiked with [1–14C] 3-phenyl propionic acid ([¹⁴C]3-phenyl propionic acid, American Radiolabeled Chemicals) to an activity per reaction of 0.167 μCi, resulting in a final substrate concentration of 40 μM and a final activity per reaction of 0.1 μCi. All monocarboxylate inhibitors were purchased from Merck (i.e., SigmaAldrich) with the exception of 6-methyl-salicylate (6MSA) that was purchased from Fluorochem. Uptake experiments were performed in 2 ml reaction tubes (Eppendorf) at 30 °C under agitation (1000 r.p.m., Thermomixer compact, Eppendorf). After 30 min, reactions were stopped by adding 900 μl stop buffer (20 mM HEPES-NaOH (pH 7.5), 150 mM NaCl) followed by centrifugation (21,000 × *g*, 4 min, room temperature). Pelleted bacteria were then washed two times with 900 μl stop buffer to remove free radioligand. Finally, bacteria were lysed using 50 μl 5% (w/v) sodium dodecylsulfate and transferred into a 96-well plate (Optiplate, PerkinElmer). 150 μl scintillation cocktail (MicroScint 40, PerkinElmer) were added to each well before measuring each well for 2 min with a scintillation counter (TopCount NXT, PerkinElmer). For data analysis Prism6 (GraphPad Software) was used.

**Expression of SfMCT in *E. coli* and membrane preparation**. For crystallization experiments, SfMCT was expressed in *E. coli* BL21(DE3)pLysS grown at 37 °C in Luria Bertani (LB) medium supplemented with antibiotics (100 μg/ml ampicillin and 36 μg/ml chloramphenicol) at 180 r.p.m. in an incubator shaker (Multitron, Infors HT). At OD₆₀₀ ~0.9, expression of SfMCT was induced by adding isopropyl-β-D-thiogalactopyranoside to a final concentration of 250 μM. After four hours, cells were harvested by centrifugation (10,000 × *g*, 6 min, 4 °C), resuspended in lysis buffer (45 mM Tris-HCl (pH 8), 450 mM NaCl, 4 °C) and pelleted again (10,000 × *g*, 25 min, 4 °C). The final cell pellet was resuspended in lysis buffer and the bacteria were lysed using an M-110P Microfluidizer (Microfluidics) operated at 1,500 bar. Low-speed centrifugation (10,000 × *g*, 10 min, 4 °C) was done to remove unlysed bacteria. Bacterial membranes were isolated by subjecting the supernatant of the low-speed centrifugation step to ultracentrifugation (200,000 × *g*, 90 min, 4 °C). The pellet was resuspended in lysis buffer, homogenized and subjected to ultracentrifugation (200,000 × *g*, 90 min, 4 °C). Finally, membranes were resuspended and homogenized in buffer (20 mM Tris-HCl (pH 8), 150 mM NaCl, 10% (v/v) glycerol) at 100 mg/ml and stored at −80 °C. Homogenization of the membranes was done using a glass teflon homogenizer (Sartorius).

**Purification of SfMCT**. Purification of SfMCT was done at 4 °C. Resuspended membranes were solubilized by gentle stirring in solubilization buffer (20 mM Tris-HCl (pH 8), 150 mM NaCl, 10% (v/v) glycerol, 4% (w/v) *n*-nonyl β-D-glucopyranoside (NG, Glycon Biochemicals GmbH)) for 2 h followed by ultracentrifugation (200,000 × *g*, 30 min, 4 °C) to remove unsolubilized material. The supernatant was diluted 1:1 with detergent-free solubilization buffer supplemented with 5 mM L-histidine. The solubilized material was then incubated with nickel-nitrilotriacetate resin (Ni-NTA; ProteinIso) (1 ml resin bed volume for solubilized membranes from one-liter expression culture) under gentle stirring for 2 h. The resin was then transferred into a column and washed with 25 column volumes of washing buffer 1 (20 mM Tris-HCl (pH 8), 150 mM NaCl, 5% (v/v) glycerol, 5 mM L-histidine, 0.4% (w/v) NG) and 25 column volumes of washing buffer 2 (20 mM Tris-HCl (pH 7), 100 mM NaCl, 0.4% (w/v) NG). SfMCT was proteolytically eluted from the column by incubating the resin with His-tagged HRV3C on a rotational shaker (~16 h)[43]. Eluted, undigested SfMCT and co-eluted HRV3C were removed by an additional Ni-NTA purification step.

**Crystallization**. Purified SfMCT was concentrated to 24 mg/ml using a 50,000 Da molecular weight cut-off concentration device (SARTORIUS Stedim Biotech, Vivaspin 2). Aggregated protein was removed by ultracentrifugation (150,000 × *g*, 30 min, 4 °C). Concentrated SfMCT was diluted to 8 mg/ml (~180 μM) using washing buffer 2 and supplemented with 1 mM monocarboxylate inhibitor from a 10 mM stock solution (10 mM monocarboxylate, 20 mM Tris-HCl (pH 7), 100 mM NaCl, 0.4% (w/v) NG). The diluted and supplemented SfMCT was incubated on ice for 1 h. The protein was crystallized in the sitting-drop vapor-diffusion method by mixing concentrated protein with reservoir solution (50 mM Tris-HCl (pH 7),

0.5 mM ZnBr$_2$, 32% (v/v) PEG-1000) using a Mosquito Crystal Robot (TTP Labtech). Crystals appeared after one day of incubation at 18 °C and reached maximal size after one week. Crystals were then collected and flash frozen in liquid ethane[44]. Before X-ray analysis, crystals were stored in liquid nitrogen.

**Data collection and structure determination**. All datasets were collected on frozen crystals at the X06SA (PXI) beamline of the Swiss Light Source (SLS; Paul Scherrer Institute, Villigen, Switzerland) using an EIGER 16M detector (Dectris). The datasets were indexed and integrated with XDS[45] and then merged using BLEND[46] of the CCP4 program suite[47] without truncation of the resolution. Scaling and averaging of symmetry-related intensities for all datasets were performed by aP_scale[48] with truncation of the data at the best high-resolution along h, k or l axis determined by AIMLESS[49]. Due to the anisotropic nature of the diffraction data the STARANISO software (http://staraniso.globalphasing.org/) was applied. This program performs an anisotropic cut-off of merged intensity data to perform Bayesian estimation of structure amplitudes and to apply an anisotropic correction to the data. The structures were obtained by molecular replacement with Phaser[50] using the SfMCT structure (PDB ID 6G9X) or the **N2C** structure (i.e., 2-naphthoate (**N2C**)-bound SfMCT structure) as search model. The final structures were obtained after multiple rounds of model building with Coot[51] and refinement with phenix.refine[52]. The following settings were used for all refinement runs: XYZ coordinates, individual B-factors, occupancies, and TLS strategies. Full data collection, processing, and refinement statistics can be found in Supplementary Table 1. Figures involving structures were prepared using PyMol (The PyMol Molecular Graphics System; Schrödinger).

**Reporting summary**. Further information on research design is available in the Nature Research Reporting Summary linked to this article.

## Data availability

Relevant data are available from the corresponding author on reasonable request. The atomic coordinates of inhibitor-bound SfMCT structures were deposited in the Protein Data Bank under the accession numbers: 6ZGR (**1OHN2C**), 6ZGS (**3PP**), 6ZGT (**N2C**), and 6ZGU (**2M3PP**).

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

## Acknowledgements

We thank Laura Baldomà (University of Barcelona, Spain) for providing the *E. coli* JA202 strain and the staff of the SLS (Paul Scherrer Institute) X06SA beamline for excellent support and advice. Financial support from the Swiss National Science Foundation SNSF (grant 310030_184980) and the NCCR TransCure is kindly acknowledged.

## Author contributions

P.D.B. and D.F. conceived and designed the experiments. P.D.B., D.K. and S.B. performed the experiments, collected and analyzed the data. P.D.B. and D.F. wrote the paper and all authors contributed to manuscript revision and approved the final version.

## Competing interests

The authors declare no competing interests.
