## [Peer Review File · Communications Chemistry]

Reviewers' comments:

Reviewer #1 (Remarks to the Author):

The manuscript COMMSCHEM-21-0123 by Dimitrios Fotiadis describes the discovery of inhibitors for SfmCT, a member of the monocarboxylate transporter family. The study is based on thiosalicylate for which a structure with SfmCT is available. Removing the sulfanyl group simplified the molecule to benzoic acid (BA) which was then the starting point for a series of five optimization steps during which each time a couple of compounds derived from the learnings in the previous round were selected and tested for inhibition of radioactive lactate uptake in *E. coli* cells expressing SfmCT. After examination of hydroxylated and methylated BA derivatives, in the next step it was found out that the combination of optimal hydroxylation and methylation had unfortunately now synergistic effect. However the positive effects of methyl groups in position C3 and C4 suggested that addition of a benzene ring might be useful to increase the potency of the inhibitors, this succeeded in a single digit micromolar inhibitor, 2-naphtoate. Next in order to increase flexibility, the ring was replaced by an sp³ hybridized alkane, this produced the first three digit nanomolar compound that could be improved further in the final step by adding a methyl group in position C2 resulting in the most potent inhibitor of the series with a K_i of 469 nM. To solidify and explain the biochemical data, 4 selected inhibitors were cocrystallized with SfmCT showing that they bind like lactate the outward open conformation and bring their carboxylate groups also in the cavity formed by L145 and F335. In addition interesting specific interactions of the 4 inhibitors are discussed that support understanding their binding behaviour and potency.

In summary this is a successful and well described MCT inhibitor optimization program that should be of interest for researchers in the field of SLC16 but also rational drug design research, it is altogether convincing based on the provided biochemical and structural data. There are a few minor points one could reflect, the optimization steps might be more easy to follow in case the compound abbreviations would be inserted into the DRCs in Fig. 1. Alternatively one might think of inserting the compound tree in Figure 1 of the supplement into the publication and adding the K_i values next to the structures. The DRCs might be moved into the supplement, since they all look very nicely like sigmoidal DRCs which however maybe does not provide a lot of novelty to the reader. The potential of these optimized inhibitors as tool compounds is mentioned in the abstract, to come to a final conclusion regarding this aspect, besides potency one should also look at specificity, i.e do these compounds also interfere with other transporters, ion channels, enzymes etc. which might then confuse the interpretation of results when using them in *in vitro* studies. In addition one might have mentioned the results from Wang, N. et al., Structural basis of human monocarboxylated transporter 1 inhibition by anti-cancer drug candidates, *Cell* 184, 2021, 370-383. At least two of these compounds are also shown to bind the monocarboxylate transporter in this case human MCT1, in the outward-open conformation in the lactate binding pocket. The high resolution crystal data in this manuscript could be briefly discussed in the context of the data generated by cryo EM structures in the Wang et al article to highlight discrepancies or common features in the binding mode.

Reviewer #2 (Remarks to the Author):

The study by Bosshart et al, reports a rational approach to designing high affinity inhibitors of the bacterial L-lactate transporter sfmCT. This transporter is a functional homologue of the human monocarboxylate transporters (SLC16) which play important roles in cancer progression. The recent

structures of sfMCT, along with the human MCT proteins has resulted in renewed interest in designing high affinity inhibitors targeting these systems in cancer. The present study is well designed and reports how the authors used their crystal structure of sfMCT to rationally design small molecule inhibitors. The authors supported the functional studies with high quality structural data and in depth analysis.

I see no reason why the present study should not be published as is. It's a very nice piece of work and I congratulate the authors.

I have only one minor comment. On page 8 line 258, the authors refer to L-lactate as having emancipated itself from obscurity as a glycolytic intermediate. I would argue that actually it is we that emancipated ourselves from our previous tunnel vision when it comes to this important metabolite. L-lactate has always been important for many cellular functions, we just didn't give it the credit!

Reviewer #3 (Remarks to the Author):

The manuscript from Bosshart et al. describes the rational development of a potent L-lactate transporter inhibitor of the MCT family. The work is based on a previous crystal structure of the L-lactate-specific SLC16 homologue SfMCT in complex with the non-transported molecule TSA, which served as a structural framework. The authors extended the initial start compound in a systematic manner and tested their ability to inhibit uptake of radiolabeled lactate in a cellular assay. They successively increased the affinity of the compound with the best inhibitors showing K_i values in the 500 nM range. The authors then determined crystal structures of SfMCT in complex with these inhibitors and confirmed their competitive nature. Due to the high quality of the electron density maps, the coordination geometry of the compounds could be unambiguously assigned in the binding pocket and was in good agreement with the functional data. This work presented here represents a comprehensive and thought-out study. The experiments were well conducted and the determined structures are high-quality models, valuable for the MCT and membrane transporter field.

I have some comments the authors could address prior to publication:

1. It is not clear where and how the authors got the compounds from? Are they all commercially available or did you synthesize some of them yourself? If this is the case, please provide information on the synthesis.
2. What is the evidence that these compounds are inhibitors and not also substrates of SfMCT?
3. The structures of human MCT1 and 2 became available recently (Wang et al., Zhang et al.). MCT1 has also been determined in two different conformations and bound to inhibitors. Would the developed inhibitors also work for the human MCTs? Is the reported binding site/mode similar as seen in the structures described here? I would encourage the authors to discuss their developed inhibitors in the context of available human MCT structures.

Reviewer #1

The manuscript COMMSCHEM-21-0123 by Dimitrios Fotiadis describes the discovery of inhibitors for SfmMCT, a member of the monocarboxylate transporter family. The study is based on thiosalicylate for which a structure with SfmMCT is available. Removing the sulfanyl group simplified the molecule to benzoic acid (BA) which was then the starting point for a series of five optimization steps during which each time a couple of compounds derived from the learnings in the previous round were selected and tested for inhibition of radioactive lactate uptake in *E. coli* cells expressing SfmMCT. After examination of hydroxylated and methylated BA derivatives, in the next step it was found out that the combination of optimal hydroxylation and methylation had unfortunately now synergistic effect. However the positive effects of methyl groups in position C3 and C4 suggested that addition of a benzene ring might be useful to increase the potency of the inhibitors, this succeeded in a single digit micromolar inhibitor, 2-naphtoate. Next in order to increase flexibility, the ring was replaced by an sp³ hybridized alkane, this produced the first three digit nanomolar compound that could be improved further in the final step by adding a methyl group in position C2 resulting in the most potent inhibitor of the series with a K_i of 469 nM. To solidify and explain the biochemical data, 4 selected inhibitors were cocrystallized with SfmMCT showing that they bind like lactate the outward open conformation and bring their carboxylate groups also in the cavity formed by L145 and F335. In addition interesting specific interactions of the 4 inhibitors are discussed that support understanding their binding behaviour and potency. In summary this is a successful and well described MCT inhibitor optimization program that should be of interest for researchers in the field of SLC16 but also rational drug design research, it is altogether convincing based on the provided biochemical and structural data.

There are a few minor points one could reflect, the optimization steps might be more easy to follow in case the compound abbreviations would be inserted into the DRCs in Fig. 1. Alternatively one might think of inserting the compound tree in Figure 1 of the supplement into the publication and adding the K_i values next to the structures. The DRCs might be moved into the supplement, since they all look very nicely like sigmoidal DRCs which however maybe does not provide a lot of novelty to the reader.

Authors: We thank the Reviewer for the positive and constructive feedback. As suggested, we inserted the compound abbreviations into the DRCs in Fig. 1 in the revised version of the manuscript. Indeed, this will make more easy to follow the optimization steps. Regarding the suggestion to eventually move the DRCs into the supplement, we have considered this possibility and prefer to keep the DRCs in the main manuscript. We hope this is acceptable for the Reviewer.

The potential of these optimized inhibitors as tool compounds is mentioned in the abstract, to come to a final conclusion regarding this aspect, besides potency one should also look at specificity, i.e do these compounds also interfere with other transporters, ion channels, enzymes etc. which might then confuse the interpretation of results when using them in *in vitro* studies.

Authors: The authors agree with the Reviewer that it would be interesting to test the effect of the described SfmMCT inhibitors on other (membrane) proteins in order to investigate their specificity. However, such a broad assay requires the set-up of a plethora of functional assays (e.g., radiolabel transport assays for membrane transporters, electrophysiological assays for ion channels and functional assays for enzymes), which is beyond the scope of this study.

In addition one might have mentioned the results from Wang, N. *et al.*, Structural basis of human monocarboxylated transporter 1 inhibition by anti-cancer drug candidates, *Cell* 184, 2021, 370-383. At least two of these compounds are also shown to bind the monocarboxylate transporter in this case human MCT1, in the outward-open conformation in the lactate binding pocket. The high resolution crystal data in this manuscript could be briefly discussed in the context of the data generated by cryo EM structures in the Wang *et al* article to highlight discrepancies or common features in the binding mode.

Authors: The authors thank the Reviewer for this suggestion/point, which was also raised by Reviewer #3. We have inserted a paragraph this regarding in the *Discussion* section of the revised version of the manuscript (page 10, top – first paragraph).

Reviewer #2

The study by Bosshart et al, reports a rational approach to designing high affinity inhibitors of the bacterial L-lactate transporter sfMCT. This transporter is a functional homologue of the human monocarboxylate transporters (SLC16) which play important roles in cancer progression. The recent structures of sfMCT, along with the human MCT proteins has resulted in renewed interest in designing high affinity inhibitors targeting these systems in cancer. The present study is well designed and reports how the authors used the authors used their crystal structure of sfMCT to rationally design small molecule inhibitors. The authors supported the functional studies will high quality structural data and in depth analysis.

I see no reason why the present study should not be published as is. It's a very nice peice of work and I congratulate the authors.

Authors: The authors thank the Reviewer for her/his positive feedback and valuable comment on our manuscript.

I have only one minor comment. On page 8 line 258, the authors refer to L-lactate as having emancipated itself from obscurity as a glycolytic intermediate. I would argue that actually it is we that emancipated ourselves from our previous tunnel vision when it comes to this important metabolite. L-lactate has always been important for many cellular functions, we just didnt give it the credit!

Authors: Indeed, an interesting point of view of the Reviewer. Accordingly, we rephrased the corresponding sentence in the revised version of the manuscript: “L-lactate is no longer solely regarded as glycolytic end-product, but is now considered an important energy source and signaling molecule in human health and disease” (page 9, first sentence of *Discussion*).

Reviewer #3

The manuscript from Bosshart et al. describes the rational development of a potent L-lactate transporter inhibitor of the MCT family. The work is based on a previous crystal structure of the L-lactate-specific SLC16 homologue SfmCT in complex with the non-transported molecule TSA, which served as a structural framework. The authors extended the initial start compound in a systematic manner and tested their ability to inhibit uptake of radiolabeled lactate in a cellular assay. They successively increased the affinity of the compound with the best inhibitors showing K_i values in the 500 nM range. The authors then determined crystal structures of SfmCT in complex with these inhibitors and confirmed their competitive nature. Due to the high quality of the electron density maps, the coordination geometry of the compounds could be unambiguously assigned in the binding pocket and was in good agreement with the functional data. This work presented here represents a comprehensive and thought-out study. The experiments were well conducted and the determined structures are high-quality models, valuable for the MCT and membrane transporter field.

I have some comments the authors could address prior to publication:

1. It is not clear where and how the authors got the compounds from? Are they all commercially available or did you synthesize some of them yourself? If this is the case, please provide information on the synthesis.

Authors: We thank the Reviewer for the positive feedback. All compounds are commercially available. We have added this information in the revised version of the manuscript (page 12, center).

2. What is the evidence that these compounds are inhibitors and not also substrates of SfmMCT?

Authors: In a previous publication, we identified SfmMCT as a proton-dependent and L-lactate-specific transporter, which does not transport the benzoate/salicylate derivative thiosalicylate (TSA) (Bosshart et al. *Nat. Commun.* 10, 2649 (2019)). The non-transported TSA is a sulfanyl-derivative of the aromatic monocarboxylate benzoate (BA), which was selected as starting compound of a rational search for molecules that potentially inhibit L-lactate transport. Motivated by the Reviewer's question, we have performed time-course transport experiments using radiolabeled salicylate (SA) and 3-phenylpropionate (3PP), which have micromolar (SA) and sub-micromolar (3PP) affinities, respectively. We have selected these compounds, because both of them have a relevant chemical structure in the rational design of our L-lactate transport inhibitor study and radiolabeled derivatives are commercially available. No SfmMCT-mediated transport was detected for SA and 3PP over a time period of 60 min. 2M3PP, the most potent compound that we have identified ($K_i = 469$ nM), is a methylated derivative of the non-transported 3PP and not commercially available as radioligand. The lack of transport of 3PP suggests that SfmMCT will also not mediate 2M3PP transport. We have inserted this new functional data in the revised version of the manuscript (Supplementary Figure 3).

3. The structures of human MCT1 and 2 became available recently (Wang et al., Zhang et al.). MCT1 has also been determined in two different conformations and bound to inhibitors. Would the developed inhibitors also work for the human MCTs? Is the reported binding site/mode similar as seen in the structures described here? I would encourage the authors to discuss their developed inhibitors in the context of available human MCT structures.

Authors: As previously shown for the SLC16 family homologue SfmMCT (Bosshart et al. *Nat. Commun.* 10, 2649 (2019)), and the cryo-EM structures of MCT1 (Wang et al. *Cell* 184, 370-383 (2021)) and MCT2 (Zhang et al. *Nat. Commun.* 11, 646 (2020)), substrates (e.g., L-lactate and pyruvate) containing a negatively-charged carboxylate group interact with the positively-charged guanidinium group of an arginine residue, which is present in TM8 of most SLC16 family members.

Comparison of residues that are important for inhibitor-binding in SfmMCT (Fig. 3) with corresponding residues in the structure of human MCT1 (Wang et al. *Cell* 184, 370-383 (2021)) indicates that there is no strong conservation, except for the conserved and functionally-essential positively-charged arginine residue in TM8 (Supplementary Figure 8). This residue is involved in ligand-binding in prokaryotic and mammalian SLC16 transporters (Bosshart et al. *Nat. Commun.* 10, 2649 (2019); Zhang et al. *Nat. Commun.* 11, 646 (2020); Wang et al. *Cell* 184, 370-383 (2021)). Alignment of the 2M3PP-bound SfmMCT structure with an outward-open cryo-EM structure of human MCT1 (Wang et al. *Cell* 184, 370-383 (2021)) suggests that the here presented SfmMCT inhibitors (i.e., 1OHN2C, N2C, 3PP, 2M3PP) might be differently oriented in human MCT1 since the distance between the N η atoms of the guanidinium group of R313 and the oxygen atoms of the carboxylate group of 2M3PP are 4.7 and 4.9 Å (Supplementary Figure 8C), which is significantly larger than observed for the inhibitor-bound SfmMCT structure (Figure 3). Furthermore, there would be clashes between the inhibitors and MCT1 side chains (Supplementary Figure 8). Nevertheless, it can be hypothesized that SfmMCT inhibitors might indicate transport-modulating effects on certain human MCTs as all compounds contain a carboxylate group.

As suggested by the Reviewer, we have discussed these points in the revised version of the manuscript (page 10, top – first paragraph).

REVIEWERS' COMMENTS:

Reviewer #3 (Remarks to the Author):

The authors reworked parts of the manuscript and replied to the comments and suggestions raised by the reviewers. Additional data have been added, certain sections have been updated and all the raised concerns have been addressed properly. This revised version is ready for acceptance.